# Comparative Venomics of the Cryptic Cone Snail Species *Virroconus ebraeus* and *Virroconus judaeus*

**DOI:** 10.3390/md20020149

**Published:** 2022-02-17

**Authors:** José Ramón Pardos-Blas, Manuel J. Tenorio, Juan Carlos G. Galindo, Rafael Zardoya

**Affiliations:** 1Departamento de Biodiversidad y Biología Evolutiva, Museo Nacional de Ciencias Naturales (MNCN-CSIC), José Gutiérrez Abascal 2, 28006 Madrid, Spain; jpblas@mncn.csic.es; 2Departamento de CMIM y Química Inorgánica-INBIO, Facultad de Ciencias, Universidad de Cádiz, 11510 Puerto Real, Spain; 3Departamento de Química Orgánica-INBIO, Facultad de Ciencias, Universidad de Cádiz, 11510 Puerto Real, Spain; juancarlos.galindo@uca.es

**Keywords:** cone snails, conotoxins, venom duct transcriptomes, venom duct proteomes, *Virroconus ebraeus*, *Virroconus judaeus*, differential expression, Cerm03, SF-mi2, insulin

## Abstract

The venom duct transcriptomes and proteomes of the cryptic cone snail species *Virroconus ebraeus* and *Virroconus judaeus* were obtained and compared. The most abundant and shared conotoxin precursor superfamilies in both species were M, O1, and O2. Additionally, three new putative conotoxin precursor superfamilies (Virro01-03) with cysteine pattern types VI/VII and XVI were identified. The most expressed conotoxin precursor superfamilies were SF-mi2 and M in *V. ebraeus*, and Cerm03 and M in *V. judaeus*. Up to 16 conotoxin precursor superfamilies and hormones were differentially expressed between both species, and clustered into two distinct sets, which could represent adaptations of each species to different diets. Finally, we predicted, with machine learning algorithms, the 3D structure model of selected venom proteins including the differentially expressed Cerm03 and SF-mi2, an insulin type 3, a *Gastridium geographus* GVIA-like conotoxin, and an ortholog to the *Pionoconus magus* ω-conotoxin MVIIA (Ziconotide).

## 1. Introduction

Cones (Gastropoda: Conidae) are venomous marine snails that produce an arsenal of different toxins (termed conotoxins) to defend against predators and to hunt worms, other snails, and fishes [1,2]. Conotoxins are produced as precursors, which include an N-terminal signal that targets the peptide at the endoplasmic reticulum, a propeptide region, which helps in the processing and folding of the peptide [3], and a mature domain, which is a relatively short peptide that establishes disulfide bonds between cysteine residues to set up the native structure of the toxin. In recent years, hormones (e.g., insulin), other peptides, and small molecules have been detected in the venom ducts, and are proposed to play an active role in the envenomation process [4,5,6]. Each time that a cone snail propels its hollow radular tooth into its prey or a predator, the venom is injected and hundreds of different peptides start blocking ion channels, as well as neurotransporters and hormone receptors. The venom composition is highly variable among the different cone species, as well as at the intraspecific and specimen levels [7,8,9]. Thus, each of the circa 1000 cone snail species known to date (MolluscaBase, accessed on 12 October 2021) provides a chance to discover a plethora of new conotoxins and potentially novel drug lead compounds. Integrated venom duct transcriptomics with proteomics and protein structure prediction constitute the field of venomics and currently represent the most comprehensive approach to study cone snail venom composition and diversity [10].

The closely related cone snails *Virroconus ebraeus* and *Virroconus judaeus* are considered cryptic species showing shells very similar in color and patterns, which are very difficult to tell apart (Figure 1). Both species can be found commonly living in sympatry in intertidal rocky habitats of the Indo-Pacific region. The species *V. ebraeus* is widely distributed in tropical waters from the east coast of Africa to the west coast of Central America, except the Red Sea [11]. Little is known about the range of distribution of *V. judaeus,* although it is considered to have a narrower distribution, comprising the Indian Ocean and the western Pacific Ocean. Although *V. judaeus* was originally described based on radular differences [12], it was not until few years ago that this species could be unequivocally differentiated from *V. ebraeus* using mitochondrial gene sequences [13]. More recently, phylogenetic relationships within *Virroconus* were analyzed based on mitochondrial and nuclear loci [14]. The mitochondrial phylogeny recovered *V. judaeus* as sister to *Virroconus coronatus* (which has very distinct shell color and pattern), whereas *V. ebraeus* was sister to *Virroconus chaldeus* [14]. Instead, the nuclear markers recovered *V. ebraeus* and *V. judaeus* as sister species and thus supported their similarities in shell morphology. The tree discordances and further statistical tests supported a past event of mitochondrial genome introgression caused by hybridization between the ancestors of *V. judaeus* and *V. coronatus* [14].

Here, we aimed at (1) uncovering the composition of *V. ebraeus* and *V. judaeus* venoms by profiling and cataloguing the venom duct transcriptomes and proteomes contents, (2) testing for significant differences between the venoms of both species and determining whether they are caused by expansion of certain conotoxin precursor superfamilies or by differential gene expression, and (3) reconstructing the native 3D structure of relevant venom peptides, which could be potential novel drug candidates.

## 2. Results and Discussion

### 2.1. Sequence Data and Assembly Quality

The venom duct transcriptomes of three specimens (OK006, OK193, and OK213) of *V. ebraeus* and three individuals (OK005, OK212, and OK214) of *V. judaeus* were sequenced (Figure 1). The Illumina RNA-seq runs produced an average of 30,553,585 and 37,885,459 paired reads for *V. ebraeus* and *V. judaeus*, respectively (see Table 1). The number of assembled transcripts for *V. ebraeus* varied between 66,731 and 68,809 (mean of 68,018), whereas this range for *V. judaeus* was larger, between 63,846 and 111,328 (mean of 90,358). BUSCO values, which evaluate the completeness and redundancy of the assemblies using universal single-copy orthologs, reported an average of 32.4% and 41.9% of complete orthologs for *V. ebraeus* and *V. judaeus,* respectively (Table 1; Appendix A). Additionally, RNA-seq data from the venom ducts of another two specimens (CebrOK014, and CebrOK016) of *V. ebraeus* and two (CebrOK033, and CebrOK035) of *V. judaeus* [14], as well as one (MAP17) of *V. ebraeus* [15], were downloaded from the SRA database and assembled. The number of raw reads from these samples varied between 10,414,401 and 16,954,014, and the number of assembled transcripts between 38,810 and 82,404 (Table 1). BUSCO values ranged between 6.9% and 30%, indicating the lower quality of some of the transcriptomes assembled from SRA (Table 1; Appendix A).

### 2.2. Identification of Virroconus Samples

Given that *V. ebraeus* and *V. judaeus* are cryptic species that can be confidently separated by means of their mitochondrial DNA sequences [13], a ML phylogeny based on the *cox1* barcoding fragment was reconstructed including all 11 samples analyzed in this paper, as well as others available in GenBank (Appendix A). According to the phylogeny, OK006, OK192, OK193, and OK213 were assigned to *V. ebraeus*, whereas OK005, OK212, OK214, and OK160 were assigned to *V. judaeus*. Only one sequence from GenBank (MG786073.1; voucher MNHN-IM-2013-50779) had a wrong species identification. This sample was initially ascribed to *V. ebraeus* (according to the information in GenBank), but was recovered within the *V. judaeus* clade in the ML tree (Appendix A). The sample location is Papua Nueva Guinea, which is a new citation for *V. judaeus* in the region, thus expanding the range of distribution of this species eastwards.

### 2.3. Conotoxin Precursor Diversity and Relative Abundance in V. ebraeus and V. judaeus

The venom duct transcriptome assemblies of the 11 *Virroconus* samples were local aligned using BLASTX against a custom-made conotoxin precursor and hormone database. The number of significant hits varied between 544 and 1734. A positive correlation was found between the initial number of transcripts and the number of hits (*p* = 2.61 × 10^−4^). The number of raw reads was also positively correlated with a high isoform/gene ratio, i.e., those samples with a higher number of reads resulted in assemblies with more transcripts but also with more isoforms (Appendix A). Each hit was aligned to the database, and manually curated to discard truncated sequences, broken open reading frames, and low coverage regions. 

After cleaning, a total number of 1275 and 1631 precursors were assigned to conotoxin superfamilies in *V. ebraeus* and *V. judaeus*, respectively (Appendix A). Since the diversity of conotoxin genes has evolved through duplication events, a reconstructed ML tree was based on signal and propeptide sequences, separating the different superfamilies into potential paralog groups, although not all superfamilies and paralogs were recovered monophyletic, such as e.g., M-3a or O2-2 (Appendix A). Conotoxin precursor transcripts from all samples were assigned to a total of 60 different superfamilies. However, seven transcripts could not be assigned to any known superfamily based on signal similarity, and thus three new superfamilies were tentatively proposed (see Section 2.5). A total of 41 and 42 transcripts were assigned to five hormone families in *V. ebraeus* and *V. judaeus,* respectively. Finally, a total of 302 transcripts were assigned to 12 proteins families related with venom performance.

The most abundant conotoxin superfamilies found in both species were M, O1, O2, T, and Cerm03 (Figure 2). The dominance in the number of transcripts of the M superfamily is in agreement with previous results reported for *V. ebraeus* [15]. The conotoxin superfamilies with more paralog groups were Conkunitzin (8), M (7), O2 (7), T (7), and Cerm03 (6) in *V. ebraeus* and Conkunitzin (8), M (8), Cerm03 (7), T (7), and O1 (6) in *V. judaeus* (Figure 2; Appendix A). The number of shared conotoxin precursors and hormones between the two species was 129 (Figure 3; Appendix A). Divergent-MKLCVVIVLL, Divergent-MRFYIGLMAA, Rimp03, Thyrostimulin-hormone-beta-5, and the new superfamily Virro01 were only present in *V. ebraeus*, whereas the A2, Cerm16, Cver05, and Cver07 and the new superfamily Virro02 were found exclusively in *V. judaeus* (Figure 3).

### 2.4. Venom Duct Proteome

Examination by UHPLC-MS of venom duct extracts from *V. ebraeus* (OK192) and *V. judaeus* (OK160) yielded 163 and 146 monoisotopic masses greater than 700 Da, respectively. Of these, 70 and 83 masses (between 707 and 7011 Da) could be assigned to sequences present in the corresponding transcriptomes of *V. ebraeus* and *V. judaeus*, respectively, after considering a variable number of identified post-translational modifications (PTMs). It must be noted that there are up to 18 naturally occurring post-translational modifications recorded in ConoServer, and some of them might account for at least some of the masses that could not be identified in the transcriptome. Some assignations were not unique, as several possible alternative sequence/PTM combinations could be consistent with the observed monoisotopic masses with identical error tolerance. Thus, 86 and 96 unique (non-redundant) sequences were compatible with the observed masses in *V. ebraeus*, and *V. judaeus*, respectively (Appendix A). Many of these sequences could be considered toxiforms, i.e., they were derived from a single peptide with a complete cysteine framework and contained variable sets of PTMs [16]. The toxiforms in *V. ebraeus* and *V. judaeus* could be ascribed to 63 and 79 unique sequences from the transcriptomes of these species, respectively.

The chromatogram of the venom duct extract of *V. ebraeus* was dominated by a very intense peak with a mass of 1619.544 Da, which was assigned to the unmodified peptide sequence CCKYPLCAAGCSCCTT in the M superfamily (Figure 4). For *V. judaeus*, the most intense peaks identified in the chromatogram (Figure 4) corresponded to modified peptides from the O1, Cerm03, M, and J superfamilies (Table 2). The prominent peak at RT 4.88 min had a mass of 2893.113 Da and was attributed to conotoxin Eb6.9 (O1-superfamily) with one N-terminal amidation (Figure 4). However, it is important to note that the difference between the observed and calculated mass of 2893.028 Da was of 0.0847 Da, greater than the tolerance limit of 0.05 Da. The sequence for conotoxin Eb6.9 was not observed in any of the transcriptomes, but sequence E_068 from CebrOK016 showed 92.6% similarity with Eb6.9 and differed only in two amino acids. The only two masses found in common to both species corresponded to Cerm03 (3256.35 Da, sequence E_207 in *V. ebraeus* OK006) and Cerm02 (2441.86 Da, sequence E_123 in E_123 CebrOK016) superfamilies, although the identification of the latter had an error mass of 0.0748 Da (i.e., greater than the 0.05 Da threshold).

The same venom extracts used for UHPLC-MS were also subjected to reduction with dithiothreitol (DTT), alkylation with iodoacetamide, and subsequent treatment with trypsin, and then analyzed by tandem nanoUHPLC-MS/MS. Since trypsin digestion produces smaller peptides, this method allowed the detection and eventual identification of sequences corresponding to peptides and proteins with elevated molecular mass, being complementary to the UHPLC-MS studies. A search against a customized database containing 20,052 venom protein sequences yielded 258 hits (unique masses) corresponding to 145 protein groups in the range of 1885 to 139,432 Da. Of these, 111 masses in 67 protein groups were common to both *V. ebraeus* and *V. judaeus*. A total of 83 masses in 44 protein groups were found only in *V. ebraeus*, and 64 masses in 34 protein groups were found only in *V. judaeus*. Protein groups were defined as those sequences from the database that gave rise upon trypsin digestion to common peptides. Hence, any of the sequences in the groups might be consistent with the observed sets of peptides detected by MS/MS analysis (Appendix A). 

The majority of the identified proteins (63–64%) corresponded to conotoxins (Figure 5). Most matched sequences obtained from the venom duct transcriptomes of *V. ebraeus* and *V. judaeus*, but a database search also yielded several records corresponding to sequences from the corresponding transcriptomes of cone snail species closely related in the phylogeny [17], such as West African species of the genera *Varioconus*, *Kalloconus*, and *Africonus*, and the Mediterranean species *Lautoconus ventricosus* [18], or to the more distantly related Indo-Pacific species of the genus *Profundiconus* [19]. In addition to conotoxins, numerous other venom proteins were identified in the proteome, including conoporins, conodipines, conkunitzins, con-ikot-ikots, protein disulfide isomerases (PDIs), peptidyl prolyl cis-trans isomerases (PPIs), ferritins, and thioredoxins, among others (Figure 5). Several prohormones and one type-3 insulin were also detected (Figure 5). Along with these proteins, a number of sequences from the so-called Metavenom Network (MVN), a cluster of housekeeping genes that co-express with venom genes in venomous amniotes, were also detected [20] (Figure 5).

The molecular mass distribution for the identified sequences in the proteomes of the venom ducts of *V. ebraeus* and *V. judaeus* is shown in Appendix A. The most abundant superfamilies represented in the proteomes of *V. ebraeus* and *V. judaeus* were M, O1, and O2, with similar percentages, and to a lesser extent T and Cerm03 (Appendix A). The most significant difference between the two proteomes was the presence of 7% conotoxins from the I2 superfamily in *V. judaeus* (Appendix A). Although conotoxin precursor transcripts of the I2 superfamily were detected in the transcriptomes of both species, no conotoxin belonging to this superfamily was observed in the proteome of *V. ebraeus*.

Venom adaptation has evolved multiple times across the animal Tree of Life [21]. Recently, a cluster of housekeeping genes that co-express with venom genes was identified in the salivary glands of venomous amniotes [20]. Due to the convergent origin of toxins along different venomous lineages, it was expected that a similar background of associated protein networks, acting as a modular unit, could be also present in cone snails. Therefore, a search for the *circa* 3000 genes of the MVN was carried out in the venom duct transcriptomes and proteomes of *Virroconus.* In fact, the presence of 1778 proteins (55% of the total MVN) shared by the transcriptomes of all *Virroconus* samples was detected. While the MVN was found to be conserved in the salivary glands of non-venomous amniotes, the venom duct of cone snails is currently considered as evolutionary derived from the mid-oesophageal gland [22]. This ontogenetic difference in the origin of the venom apparatus together with the highly distant position of both groups within the animal Tree of Life, could partially explain the absence of some members of the amniote MVN in cone snails. The search results for the proteome were less successful, as only 16 and 28 MVN proteins were detected in *V. ebraeus* and *V. judaeus*, respectively (all proteins found in *V. ebraeus* were shared with *V. judaeus*). The low number of samples used in the proteome analysis together with an experimental design to capture small peptides could in part explain the poor MVN yields obtained in the proteome analyses. The more the venom ducts of cone snails and other venomous invertebrates are analyzed, the more it will become clear how likely the convergent co-option of MVN proteins out of amniotes has been.

### 2.5. New Putative Conotoxin Precursor Superfamilies Identified in Virroconus

The signal sequence of seven transcripts did not match (>70% identity) any of the known conotoxin precursor superfamilies, and thus three new putative superfamilies were proposed. Virro01 has a signal sequence (MLKMPVLLLTILLLLLMATA) most similar to the Y (60.0% identity) and M (63.2% identity) superfamilies and a cysteine framework type VI/VII, which is widespread among conotoxin precursor superfamilies. Virro02 has a signal sequence (MGKLTKVLLFAAVLMLTQIMDQGEG) most similar to the O2 superfamily (65.2%) and a cysteine framework type XVI, which has been found in the Q and M superfamilies. Virro03 has a signal sequence (MMNLVSMTIVLLLLAQCQLFTA) most similar to the O1 superfamily (59.2% identity) and has a cysteine framework type VI/VII. These new superfamilies showed minimal expression values, except for Virro02, which had 1228 TPMs (transcripts per million) in OK214. Moreover, Virro02 peptides were identified in the proteomes of both *V. ebraeus* and *V. judaeus*. Further research will be needed to validate the possible biological activity of these putative new superfamilies.

### 2.6. Differences in Conotoxin Precursor Expression between V. ebraeus and V. judaeus

The expression of the different conotoxin precursor superfamilies and hormones was compared between *V. ebraeus* and *V. judaeus* to find significant differences between these cryptic species and associate them to potential diet differences. First, relative expression was analyzed by normalizing the count for each of the transcripts as TPMs, and by aggregating the values of all sequences belonging to the same superfamily. The most expressed conotoxin precursor superfamilies in *V. ebraeus* were SF-mi2, M, O1, B2, and F, with the SF-mi2 and M superfamilies accounting for up to 49.4% of all expression. This pattern is in agreement with most expressed superfamilies previously reported for *V. ebraeus* [15]. The most expressed superfamilies in *V. judaeus* were Cerm03, M, O2, I2, and O1, with the Cerm03 and M superfamilies accounting for up to 49% of the expression.

A second analysis mapped clean reads against conotoxin precursor and hormone transcripts of the 11 transcriptomes (Appendix A). After adding up the number of TPMs within each superfamily, a principal component analysis (PCA) was carried out (Appendix A). The pattern of expression was clearly distinct between the two species along the PC1 component, positive in *V. ebraeus* and negative for *V. judaeus* (Appendix A). The only exception was the striking position of *V. judaeus* CebrOK035, which was found almost in the 0 value, slightly negative of the PC1, and separated from the other two groups by the negative load of the PC2 (Appendix A). The O2 and M superfamilies were found to be the main contributors to this effect. 

Additionally, a differential expression analysis of conotoxin precursor and superfamilies and hormones between the 11 venom ducts transcriptomes was conducted. The Cerm10, Con-ikot-ikot, F, I3, Pmag02, Q, Rimp01, SF-mi2, and Tpra06 superfamilies in *V. ebraeus,* and Cerm03, Conkunitzin, Ggeo03, H, I2, O2, and T in *V. judaeus* were found to be differentially expressed (Appendix A). The analysis clearly separated the samples of both species into two different expression groups. However, since samples were generated in different sequencing platforms and had variable sequencing depths and read length, the analysis was repeated using only the six transcriptomes newly sequenced in this work, which were all generated and processed in the same manner (Figure 6). The new analysis also recovered both species as different expression clusters and reported a total of 13 differentially expressed superfamilies (all previously detected except the Cerm10, H, and Tpra06 superfamilies; Figure 6).

Up to seven conotoxin precursor superfamilies (Cerm03, Con-ikot-ikot, F, I3, O2, Pmag02, and SF-mi2) added up more than 10,000 TPMs (when both species were considered) (Appendix A). Of these, the I3 and F superfamilies were found 41 and 32 times more expressed in *V. ebraeus* than in *V. judaeus*, respectively, whereas the O2 superfamily was almost nine times more expressed in *V. judaeus* than in *V. ebraeus* (Appendix A). Nonetheless, the expression values of two superfamilies stood out from any other: the loadings for PC1 clearly indicated that the Sf-mi2 and Cerm03 superfamilies were overexpressed in *V. ebraeus* and *V. judaeus,* respectively (Appendix A).

The Cerm03 superfamily was found 19 times more expressed in *V. judaeus* than in *V. ebraeus* with a total number of 352,046 TPMs (Appendix A). Cerm03 was classified as a new superfamily in *Chelyconus ermineus* [7]. The SF-mi2 superfamily was found 265 times more expressed in *V. ebraeus* than in *V. judaeus* with a total number of 300,214 TPMs (Appendix A). This superfamily was originally identified in the venom duct of *Rhizoconus miles* [23] and later reassigned to the G2 superfamily based on the 3D structure, which had two β-hairpins that resembled the domain of granulin [24]. More recently, the superfamily was also found in *Pionoconus magus* and named Rmil02 [8]. The three names correspond to the same superfamily, and we use here SF-mi2 as the valid name, given that it is the oldest available. The recent and successful application of transcriptomics to the identification of conotoxin precursors is rapidly increasing the number of new superfamilies discovered, and the example mentioned above highlights that a broad consensus around conotoxin classification and nomenclature is urgently needed. 

Both the Cerm03 and SF-mi2 superfamilies showed prominent peaks in the proteome analyses (Figure 4), which is consistent with their high expression levels. One of the most intense peaks in the chromatogram of the venom duct extract of *V. judaeus* OK160 at RT = 5.36 min, with a monoisotopic mass of 3256.352 Da, was attributed to a conopeptide of the Cerm03 superfamily (OK005_J_237; Figure 4). Similarly, a peak at RT = 0.97 min with a mass of 3026.162 Da in the chromatogram of the venom duct extract of *V. ebraeus* OK192 was identified as a member of the SF-mi2 superfamily (OK006_E_244; Figure 4).

Altogether, the above-mentioned expression analyses indicate that *V. ebraeus* and *V. judaeus* have similar venom toolkits in terms of the relative abundance of conotoxin precursor superfamilies, but show marked differential expression. Most recent phylogenomic analyses indicate that *V. ebraeus* and *V. judaeus* are closely related sister species, and that the latter has an introgressed mitochondrial genome from *V. coronatus* [14]. Hence, the rather similar composition of their venoms at the superfamily level (Figure 2) as previously reported in closely related species within genus *Turriconus* [25] or within genera *Africonus* and *Varioconus* [9]. However, it is possible to distinguish two sets (Figure 5 and Appendix A) of conotoxin precursor superfamilies differentially expressed in *V. ebraeus* (Cerm10, Con-ikot-ikot, F, I3, Pmag02, Q, Rimp01, SF-mi2, and Tpra06) and *V. judaeus* (Cerm03, Conkunitzin, Ggeo03, H, I2, O2, and T). In *Turriconus*, a similar pattern was reported as the O1d and P superfamilies showed a higher relative expression in *Turriconus andremenezi* and *Turriconus praecellens*, respectively [25]. The significant differences in conotoxin gene expression may represent adaptations to different diets. The impact of specialized feeding strategies in the isolation of populations has been proposed as a mechanism to explain the co-occurrence and evolution of sister marine species living in sympatry [26,27]. Both studied *Virroconus* species are vermivorous and prey on polychaetes. It is known that juvenile individuals of both species prey mainly on worms of the family Syllidae [13]. However, it has been shown that the adult individuals of *V. ebraeus* prey on Nereididae and Eunicidae, whereas adults of *V. judaeus* prefer preying on Capitellidae [13]. Eunicidae and Nereididae are able to move freely, whereas Capitellidae are mainly found in tubes in the sediment as deposit feeders. The diet shifts during the development in these species are coupled with a change in the morphology of the radular tooth during the ontogeny of both species, as has been also depicted for *Pionoconus magus* [28], and it is potentially occurring in other cone snails with different prey preferences [29]. Altogether, our results suggest that *V. ebraeus* and *V. judaeus* express different conotoxin superfamilies, which could contribute, together with radular differences, to a possible specific specialization in preying on different polychaetes families during the adult stages.

### 2.7. Molecular Modeling

The recent application of artificial intelligence (neural networks) to accurately predict the tertiary structure of proteins and peptides directly from their amino acid sequences has been an important landmark on the way to solving the protein-folding problem, i.e., how the amino acid sequence of a protein dictates its three-dimensional atomic structure [30]. The Local Distance Difference Test (expressed as pLDDT) defines the quality of the modeled structure: values of 80 or above indicate a highly accurate modeled structure, although this value may vary along the amino acid chain, alternating regions accurately modeled with others of low quality, often disordered (i.e., chains connecting two protein domains). This approach was used to model several interesting mature conotoxins and hormones identified during the analyses of the venom duct transcriptomes and proteomes of *Virroconus* (Appendix A).

#### 2.7.1. OK005_J_237

The 28-AA peptide OK005_J_237_Cerm03 was highly expressed in the transcriptome of *V. judaeus* and produced one of the most intense peaks in the corresponding chromatogram. Its structure was modeled with a moderate to high degree of accuracy (pLDDT = 77.40, Appendix A). This conopeptide contains four cysteine residues with a framework XIV, and with a disulfide connectivity I-III, and II-IV. The folded peptide is bound to a relatively long (11 AA) cysteine-free amino acid tail, which was not modeled so accurately. The possible molecular targets of this peptide are unknown.

#### 2.7.2. OK006_E_244

The structure of the peptide OK006_E_244_Sf-mi2, which was highly expressed in the transcriptome of *V. ebraeus*, was modeled with a high degree of accuracy (pLDDT = 91.14; Appendix A). This 28-AA conopeptide contains eight cysteine residues with a framework XXVII, and with a disulfide connectivity I-IV, II-VI, III-VII, and V-VIII. This disulfide arrangement is identical to that recently reported for the conotoxin Φ-MiXXVIIA of *Rhizoconus miles* [23]. Furthermore, the modeled structure of OK006_E_244_Sf-mi2 and that of conotoxin Φ-MiXXVIIA displayed essentially the same folding, as shown by the superimposition of both structures (Figure 7A). This folding resembles the N-terminal domain of granulin and has been shown to be connected to anti-apoptopic activity [24]. It is therefore likely that this peptide might exhibit similar bioactive properties.

#### 2.7.3. OK193_E_124

This 25-AA peptide of the O1 superfamily has a sequence with a 72% identity to that of *P. magus* ω-conotoxin MVIIA (Ziconotide), so a similar structure could be anticipated. In fact, the accurately modeled structure of this peptide (Appendix A) showed a cysteine framework VI/VII with disulfide connectivity I-IV, II-V, and III-VI, as expected. The superposition of the model with the experimentally determined structure of conotoxin MVIIA [31] showed an essentially identical folding with the same arrangement of disulfide bridges (Figure 7B). This structural similarity most likely suggests an analog biological activity, targeting calcium ion channels. Hence, peptide OK193_E_124 (O1) can be considered a putative member of the ω pharmacological family and might be a potentially interesting drug lead compound. However, this peptide was only low expressed in the transcriptome (20 TPM) and was not present in the proteome of the specimens studied.

#### 2.7.4. OK213_E_290

The insulin-related peptide OK213_E_290_insulin has 191 AA. It was modeled with lower quality compared to other models in this work (pLDDT = 56.52). However, this is an average value, and the structure showed a series of peptide chains linking well-modeled domains (Appendix A). These domains compared very well with the structure of the recently reported insulin Con-Ins G1, found in the cone snail *Gastridium geographus* [32] (Figure 7C). This insulin is the smallest known insulin found in nature so far, with a tertiary structure similar to that of human insulin [33]. The insulin-related peptide OK213_E_290 was also identified in the proteome of the venom duct extract of *V. ebraeus* OK192.

#### 2.7.5. Eb6.9

The sequence corresponding to the canonic conotoxin Eb6.9 (containing a N-terminal amidation) was identified in the proteome of the venom duct extract of *V. judaeus* OK160 and in the transcriptome of *V. ebraeus* CebrOK016_E_068. The structure of this 28-AA peptide of the O1 superfamily was modeled with a high degree of accuracy (pLDDT = 92.70; Appendix A). This conopeptide contains six cysteine residues with a framework VI/VII, and a disulfide connectivity I-IV, II-V, and III-VI, (Appendix A). Conotoxin Eb6.9 has a structure very similar to that of ω-conotoxin GVIA of *G. geographus* [34], as inferred by the superposition of the models of both structures (Figure 7D). The bioactivity of ω-conotoxin GVIA has been thoroughly studied, and it is a well-known N-type calcium channel blocker [36]. Given the structural affinities of conotoxin Eb6.9 and GVIA, it seems reasonable to assume that the former also has calcium channel blocker capabilities, but this remains to be demonstrated.

#### 2.7.6. CebrOK033_J_232

Sequences corresponding to peptides of the new superfamilies Virro01, Virro02, and Virro03 were modeled. Unfortunately, for two of the superfamilies (Virro01 and Virro02), the structure modeling using AlphaFold rendered non-significant results (average pLDDT < 60). On the other hand, for the sequence CebrOK033_J_232, a member of the new superfamily Virro03, a high-quality model (average pLDDT = 89) was obtained (Appendix A). This 37-AA conopeptide contains six cysteine residues with a framework VI/VII, and a disulfide connectivity I-IV, II-V, and III-VI. The resulting structure was very similar to that of μ-theraphotoxin-Pn3a from the spider *Pamphobeteus nigricolor* (giant blue bloom tarantula) [35], as inferred by the superposition of models of both structures (Figure 7E). μ-Theraphotoxin-Pn3a is a potent inhibitor of voltage-gated sodium channel Na_V_1.7. Given the structural resemblances between this peptide and CebrOK033_J_232, a similar pharmacological function for the member of the Virro03 superfamily might be anticipated, although this is yet to be demonstrated. In this context, it has been shown that the presence of the positively charged aminoacids K22 and K24 was considered crucial for Pn3a activity [37]. In CebrOK033_J_232, these positions are occupied by different amino acids, namely arginine (R22) and serine (S24). Replacement of K22 by arginine in Pn3a reduces the activity, while position 24 seems more tolerant to modifications. This might indicate a lower activity of CebrOK033_J_232 compared to Pn3a for Nav1.7 inhibition or modifications in the putative targets of CebrOK033_J_232. Members of the Virro03 superfamily were not detected in the proteomes of *V. ebraeus* or *V. judaeus*, at variance with members of the Virro02 superfamily, which were present in both species (Appendix A).

## 3. Material and Methods

### 3.1. Sampling and Venom Duct Processing

Four specimens of *V. ebraeus* (OK006, OK192, OK193, and OK213) and *V. judaeus* (OK006, OK160, OK212, and OK214) were collected on Okinawa and Ishigaki islands (Japan) in 2017 (Table 1). Specimens OK192 and OK160 were used for proteomic analysis, whereas the others were set for transcriptomic analysis. Once the animals were in resting state, they were extracted from their shells to obtain the whole venom duct, which was stabilized in 1 mL RNAlater (Thermo Fisher Scientific, Waltham, MA, USA) at −20 °C. 

### 3.2. Transcriptomic Analysis of the V. ebraeus and V. judaeus Venom Duct

Following a well-established and efficient procedure [7], each venom duct was ground in 300 µL of TRIzol (Thermo Fisher Scientific, Waltham, MA, USA) and mixed with 60 µL of chloroform. After centrifugation, the aqueous phase was recovered and RNA precipitated in one volume of isopropanol and incubated overnight at −80 °C. The Direct-zol RNA miniprep kit (Zymo Research, Irvine, CA, USA) was used to purify total RNA (5–15 µg) following the manufacturer’s instructions.

Dual-indexed cDNA libraries (307–345 bp insert size) for each sample were constructed after isolation of mRNA using the TruSeq RNA Library Prep Kit v2 (Illumina, San Diego, CA, USA) and following the manufacturer’s instructions. The quality of the libraries was analyzed with the 4200 TapeStation and the High Sensitivity D1000 ScreenTape assay (Agilent Technologies Inc., Santa Clara, CA, USA); libraries were quantified using real-time PCR in a LightCycler 480 (Roche Molecular Systems Inc., Pleasanton, CA, USA). The pool of indexed libraries (including samples from other projects) was loaded into different lanes and sequenced by paired-end sequencing (2 × 100 bp) in an Illumina HiSeq2500 (two flow cells) following standard procedures at Sistemas Genómicos (Valencia, Spain).

#### 3.2.1. Transcriptome Assembly

Together with the six newly generated RNA-Seq data sets, raw reads from two *V. ebraeus* (CebrOK014, and CebrOK016) and two *V. judaeus* (CebrOK033, and CebrOK035) RNA-Seq data sets [14], as well as from one *V. ebraeus* (MAP17) RNA-Seq data set [15], available in the SRA database (https://www.ncbi.nlm.nih.gov/sra, accessed on 10 February 2021) were assembled.

Quality was checked using FASTQC [38] with default parameters. Trinity v2.9.1 [39] was run to reconstruct a de novo transcriptome assembly for each of the 11 samples, with default parameters, and Trimmomatic activated with default parameters. To check the quality of the assembly, BUSCO v4.0.6 [40] was used with default parameters and metazoaodb_10. The FASTQC and BUSCO reports were merged into a single report (Appendix A) using MultiQC [41].

#### 3.2.2. Conotoxin Precursor Prediction and Classification

A custom protein reference database was constructed by merging the conotoxin precursors available in Conoserver [42] together with the results of searches using “conotoxin” as a query term in GenBank and Uniprot [43] (both accessed on 22 February 2021), as well as conotoxin precursors identified in *Profundiconus* [19], *Cylinder gloriamaris* [44], West African cone species [9], and *Lautoconus ventricosus* [18]. Misidentified sequences from public databases were deleted from the database (see Appendix A in [9]). Redundancy was reduced with CD-HIT v4.7 (-c 0.99) [45] to produce the final conotoxin precursor database.

BLASTX (1 × 10^−6^) searches of assembled transcripts from each sample against the conotoxin database were performed. Query hits were translated into the six open reading frames and aligned to the database hits in order to check broken ORFs, chimeric transcripts, or dubious results. Resulting sequences were classified into different conotoxin precursor superfamilies using the Conoprec tool [42] and searches using BLASTP (1 × 10^−6^) to the NCBI. Conotoxin precursor superfamilies corresponded to transcript sequences that showed more than 70% of identity in the signal region. Further classification into potential paralog groups was based on a maximum likelihood (ML) tree of the signal and propeptide regions (Appendix A) that was reconstructed with PhyML v3.0 [46] in the ATGC platform (http://www.atgc-montpellier.fr/, accessed on 6 April 2021). The ML tree was visualized using FigTree (tree.bio.ed.ac.uk/software/figtree/). Transcripts whose sequences represented less than 55% of a canonical sequence of the group were deleted. Clean reads were mapped to the extracted coding sequences (cds) with Bowtie2 v2.2.3 [47] in order to check the coverage of the cds. Visualization was performed in IGV [48]. Finally, a TBLASTN of the cds against the GenBank database was performed to check for potential artifacts and false conotoxin precursors.

#### 3.2.3. Shared Conotoxins between *V. ebraeus* and *V. judaeus*

Conotoxin precursors and hormones from all samples were clustered using CD-HIT v4.7 (cd-hit-c 1.00-d 0-sc 1). Output clusters were manually transformed into intersect files for *V. ebraeus* and *V. judaeus*. Venn diagrams were plotted with the R package Venn (https://cran.r-project.org/web/packages/venn/index.html, accessed on 30 April 2021), and *V. ebraeus* and *V. judaeus* intersects were plotted with UpSetR [49].

### 3.3. Virroconus Species Identification

Due to the highly similar shell morphology of the *V. ebraeus* and *V. judaeus* specimens, we carried out DNA comparisons between samples. The barcoding mitochondrial *cox1* gene fragment of 594 bp was isolated from the 11 assembled transcriptomes by BLASTX (E-value 1 × 10^−6^) using the *cox1* sequence from *Spuriconus spurius* as the query. Additionally, a barcoding *cox1* fragment was PCR amplified (94 °C for 60 s/45 cycles at 94 °C for 30 s, annealing at 45 °C for 30 s, extension at 70 °C for 90 s/final extension at 72 °C for 5 min) using the universal COI primers for invertebrates [50] from DNA extracted from OK160 and OK192 and sequenced. All newly determined sequences were aligned with the ortholog sequences of *V. ebraeus* and *V. judaeus* downloaded from GenBank using TranslatorX [51]. The resulting matrix was used as input for PhyML v3.0 [46] in the ATGC platform (http://www.atgc-montpellier.fr/, accessed on 6 April 2021) to reconstruct an ML tree with 1000 bootstraps. The ML tree was visualized using FigTree (tree.bio.ed.ac.uk/software/figtree/).

### 3.4. Differential Expression Analyses

Clean reads from trimmomatic were aligned to the cds of conotoxin precursors and hormones using the pseudo-aligner Kallisto v0.46.0 [52]. Raw estimated counts of each transcript belonging to the same superfamily were added up to create a matrix (Appendix A). Superfamilies that had 0 counts in any of the samples were not included in the expression analysis. EBSeq [53] was used to estimate differential expression between *V. ebraeus* and *V. judaeus* using each of the samples as a biological replicate. A superfamily was considered differentially expressed when the posterior probability (PPDE) was ≥0.95. Heatmaps with the differentially expressed superfamilies were based on transcripts per million (TPMs) mapped by Kallisto and plotted using the R package pheatmap (https://cran.r-project.org/web/packages/pheatmap/index.html, accessed on 15 May 2021) with the Z-score (number of standard deviations, respect to the mean between the values of each superfamily) to normalize data.

### 3.5. Proteomic Analysis of the V. ebraeus and V. judaeus Venom Ducts

#### 3.5.1. Venom Extraction

The venom ducts of one specimen of *V. ebraeus* (OK192) and one of *V. judaeus*, (OK160), both preserved in RNAlater, were freeze dried and cut into small pieces (less than 1 mm). The lyophilized venom ducts were extracted with 1.5 mL of 0.1% formic acid in water. The samples were centrifuged at 16,100× *g* for 15 min. The supernatant fractions were separated, and the pellets were re-extracted and centrifuged two more times, repeating the same procedure. The combined extracts were filtered, and the solutions of crude venom were stored at −20 °C.

#### 3.5.2. RP-UHPLC/MS Analysis

Reverse phase ultra-high-performance liquid chromatography (RP-UHPLC) was carried out at the Servicio Central de Ciencia y Tecnologia (SCCyT) of the University of Cadiz on an ACQUITY H-Class UHPLC system (Waters, Milford, MA, USA), with a binary solvent system and an automatic sample manager equipped with an ACQUITY UPLCr BEH C18 (2.1 × 50 mm, 1.7 µm) column, running at 50 °C. The mobile phases consisted of eluent A (0.1% formic acid in water, *v*/*v*) and eluent B (0.1% formic acid in acetonitrile, *v*/*v*). These phases were delivered at a flow rate of 0.6 mL/min by using a linear gradient program as follows: 0–14 min, 95–5%A; 14–16 min, 5–95% A; and 16–18 min, 95–5% A. The injection volume was 2 μL. The UHPLC was coupled to a XEVO-G2-S QTOF quadrupole time-of-flight tandem mass spectrometer equipped with an electrospray (ESI) source. The operating parameters in ESI were set as follows in positive mode: sample cone voltage of 20 V, source temperature of 120 °C, cone gas flow of 10 L/h, and desolvation gas flow of 850 L/h. Acquisitions were carried out over the range of 100 to 3000 Da. The *m*/*z* Leucine-enkephalin (*m*/*z* 556.2771 in positive ion mode) was used as the external reference of LockSpray infused at a constant flow of 5 μL/min. Data acquisition and visualization was performed using MassLynx 4.1 software (Waters, Manchester, UK) in positive-ion mode. 

From the LC/MS analysis, a list of monoisotopic masses was obtained for each of the species analyzed. Duplicated masses were removed by using the corresponding tool in Conoserver [42], setting the allowed difference between identical masses in 0.05 Da. The resulting monoisotopic masses were annotated by means of the ConoMass tool, also in ConoServer. In a first step, a FASTA file containing sequences of mature peptides with <90 amino acids (extracted from the *Virroconus* transcriptomes) was uploaded to determine posttranscriptional modifications (PTMs). Apart from disulfide bonds, the following PTMs were selected: N-terminal amidation, pyroglutamylation, glutamate carboxylation, proline and valine hydroxylation, and tryptophane bromination. In a second step, peptides were identified in the monoisotopic mass list by matching with the calculated PTM differential mass file. The error between experimental and computed masses was set to 0.1 Da, and a threshold of 0.05 Da was subsequently applied. 

#### 3.5.3. Shotgun Proteomics (LC-MS/MS)

The lyophilized extract (5 mg) was resuspended in 100 µL of RapiGest SF (Waters, Milford, MA, USA) at 0.2% in 50 mM ammonium bicarbonate, and 10 µL of this solution were reduced, alkylated, and digested with trypsin as follows: 2.5 µL of DTT were added at a concentration of 50 mM (in 50 mM ammonium bicarbonate) and incubated for 30 min at 60 °C. Subsequently, 3 µL of Iodoacetamide were added at a concentration of 100 mM (in 50 mM ammonium bicarbonate) and incubated for another 30 min at room temperature in the dark. After reduction and alkylation, trypsin (Promega, Madison, WI, USA) was added in two steps: first, 1.25 µg of trypsin were added and incubated at 37 °C for 2 h; then the same amount of trypsin was added and incubated for another 15 h. The final volume of the digestion was brought to 100 µL with milliQ water. The result of the digestion was diluted 1:10 with formic acid 0.1%, and 2 µL of each digest were analyzed by liquid chromatography coupled to mass spectrometry at the Research Core Facilities in Biomedicine (SC-IBM), University of Cadiz, using a nanoElute nanochromatograph (Bruker) and a quadrupole-time-of-flight hybrid mass spectrometer (Q-TOF) Tims-TOF Pro (Bruker, Billerica, MA, USA). Peptides were eluted in a 60 min gradient from 3–35%, A being water with 0.1% formic acid, and B acetonitrile with 0.1% formic acid. The chromatography flow was 300 nL/min. The column used was ionopticks C18 (25 cm × 75 μm id, 1.6 μm; Fitzroy, Australia).

Similar to the peptides eluted from the chromatography, they were ionized in a Captive electrospray source (Bruker, Billerica, MA, USA) at 1500 V, and analyzed by mass spectrometry with a DDA-PASEF method, where the peptides are isolated and fragmented according to their mass/charge and its ion mobility values. The DDA-PASEF method consisted of 10 MS/MS PASEF scans per topN acquisition cycle, with an accumulation time of 100 ms and a ramp of 100 ms. The MS and MS/MS spectra were acquired in a *m*/*z* range of 100 to 1700 and in an ion mobility range (1/K_0_) of 0.60 to 1.60 V s/cm^2^, selecting the precursor ions for the MS/MS PASEF scans from a previous TIMS-MS scan. The collision energy was programmed as a function of ion mobility, following a line from 20 eV for 1/K_0_ of 0.6 to 59 eV for 1/K_0_ of 1.6. The TIMS elution voltage was linearly calibrated to obtain 1/K_0_ coefficients using three ions (*m*/*z* 622, 922, and 1222) from the ESI-L Tuning Mix (Agilent, Santa Clara, CA, USA).

#### 3.5.4. Bioinformatic Integration of Proteomic and Transcriptomic Data

PEAKS Pro Studio software (Bioinformatics solutions, Waterloo, ON, Canada) was used to match MS/MS spectra obtained from proteomic analysis of *Virroconus* species venom. MS spectra of peptides were elucidated based on a database containing all conotoxin sequences in Conoserver, mature sequences of all published cone snail venom duct transcriptomes up to date (including those of *Virroconus* species determined in the present work and in [14]), sequences searched in GenBank and UniProt databases using the term “conotoxin”, plus a series of highly conserved protein sequences recently found in venomous amniotes (MVN, [20]). This database contained a total of 20,052 protein sequences. 

In order to assess the presence of MVN proteins [20], the Id genes from the turquoise module (https://agneeshbarua.github.io/Metavenom/#Gene_network_analysis, accessed on 14 July 2021) were used to download the corresponding proteins from GenBank. A TblastN (1 × 10^−6^) of each of the 11 transcriptomes against the MVN database was performed. Hits of each transcriptome were collected (≥50% identity and an E-value ≥ 1 × 10^−6^). The headers of the snake sequences were isolated using bash scripting and compared to each other to obtain the shared hits between all the *Virroconus* transcriptomes.

Carbamidomethylation of cysteine was set as fixed modification, while methionine oxidation, amidation of C-terminus, proline oxidation to pyroglutamic acid, and pyroglutamic from glutamic acid and glutamine were set as variable modifications, with the parameter for the maximum missed cleavages at 3 for trypsin digestion. Parent mass and fragment mass error tolerance were set at 15 ppm and 0.015 Da, respectively. A false discovery rate (FDR) of 1% and unique peptide 1 were used for filtering out inaccurate proteins. A-10lgP > 80 was used to estimate whether the detected proteins were identified by enough reliable peptides MS/MS spectra.

### 3.6. Molecular Modeling

Peptide 3D structures were predicted using multiple sequence alignments (MSA) generated through an MMseqs2 application interface as implemented in ColabFold [54], which uses the recently released AlphaFold2 source code [30]. Sequences were entered into the ColabFold notebook (https://colab.research.google.com/github/sokrypton/ColabFold/blob/main/beta/AlphaFold2_advanced.ipynb, accessed on 5 September 2021), with the following advanced features: msa_method = MMSeq2, num_models = 5, num_relax = Top1. The quality of the best model was assessed using the mean Local Distance Difference Test (pLDDT). A pLDDT score ≥60 was considered a reasonable model, and scores >80 indicated a very accurate model. UCSF Chimera 1.15 [55] was used for viewing and manipulating the molecular graphics (pdb files) for the modeled structures.

## Figures and Tables

**Figure 1 marinedrugs-20-00149-f001:**
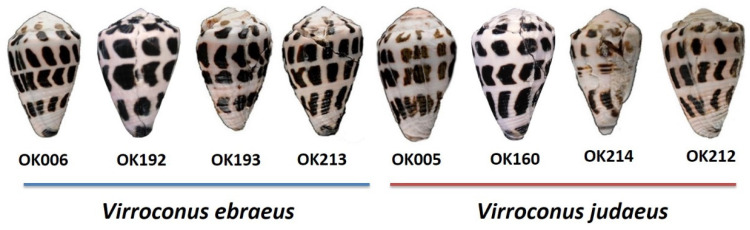
Dorsal view of the shell of the new specimens of *V. ebraeus* and *V. judaeus* included in this work. The sizes of the pictures are not scaled.

**Figure 2 marinedrugs-20-00149-f002:**
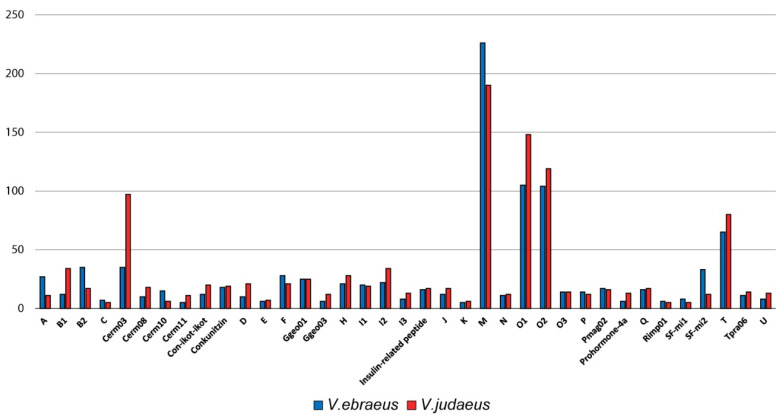
Histogram showing the relative abundance of conotoxin precursor superfamilies and hormones in *V. ebraeus* (blue) and *V. judaeus* (red). Y-axis indicates the number of different transcripts; duplicated sequences for the same species were collapsed. Superfamilies with fewer than five members for one or both species are not shown.

**Figure 3 marinedrugs-20-00149-f003:**
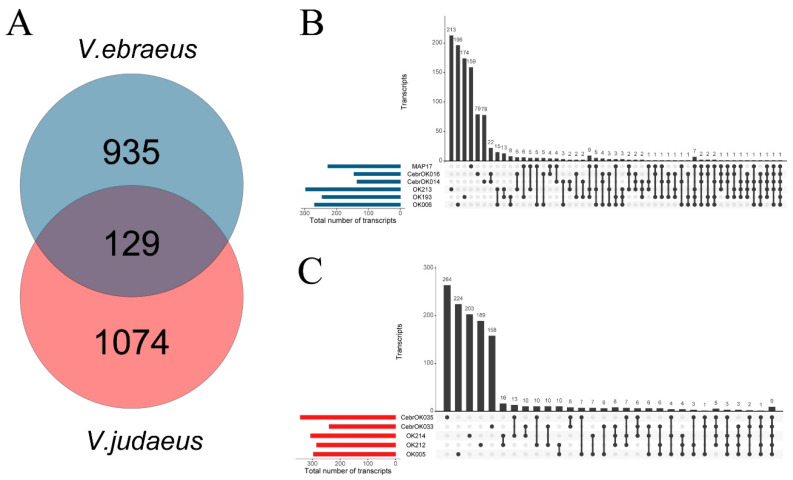
Conotoxin precursor and hormone transcripts of *V. ebraeus* (blue) and *V. judaeus* (red). (**A**) Venn plot showing shared transcripts between the six specimens of *V. ebraeus* and the five samples of *V. judaeus*. Shared transcripts among samples within *V. ebraeus* (**B**) and *V. judaeus* (**C**) are depicted. Colored bars indicate the total number of precursors of each sample, whereas the linked dots indicate the shared transcripts between samples. Y axis indicates the number of transcripts.

**Figure 4 marinedrugs-20-00149-f004:**
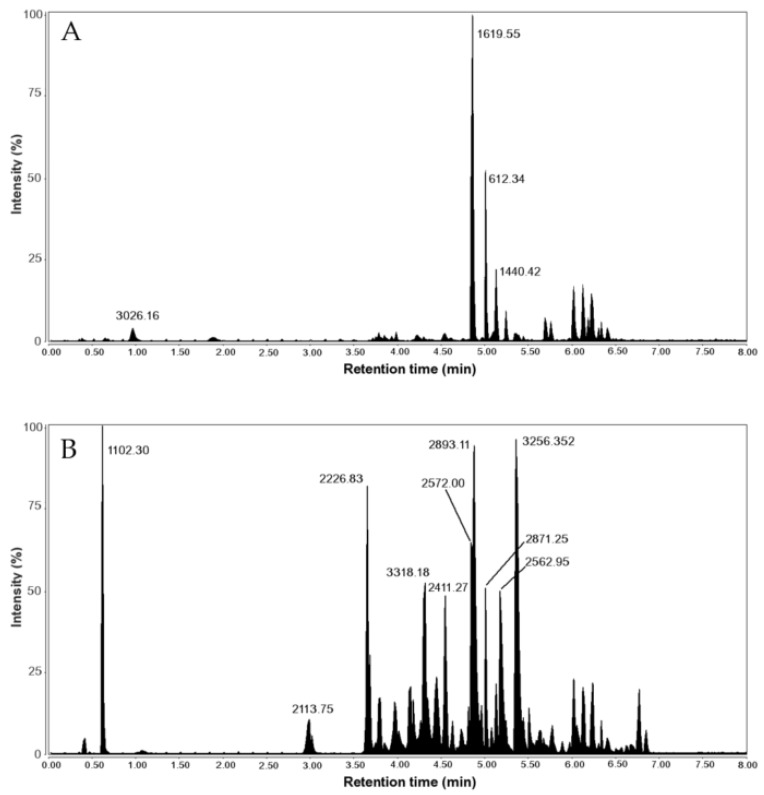
UHPLC-MS chromatogram (total ion current trace) of venom duct extracts of *V. ebraeus* OK192 (**A**) and *V. judaeus* OK160 (**B**) with selected peaks labeled with their corresponding monoisotopic masses (Da).

**Figure 5 marinedrugs-20-00149-f005:**
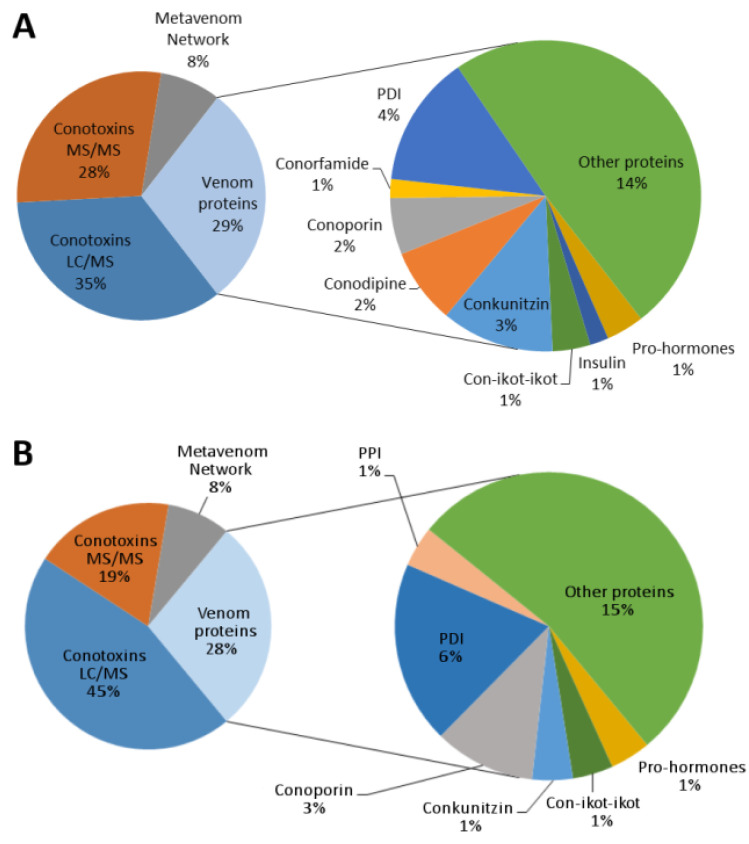
Proteome composition of the venom duct extracts of *V. ebraeus* OK192 (**A**) and *V. judaeus* OK160 (**B**). PDI, protein disulfide isomerase; PPI, peptidyl prolyl cis-trans isomerase.

**Figure 6 marinedrugs-20-00149-f006:**
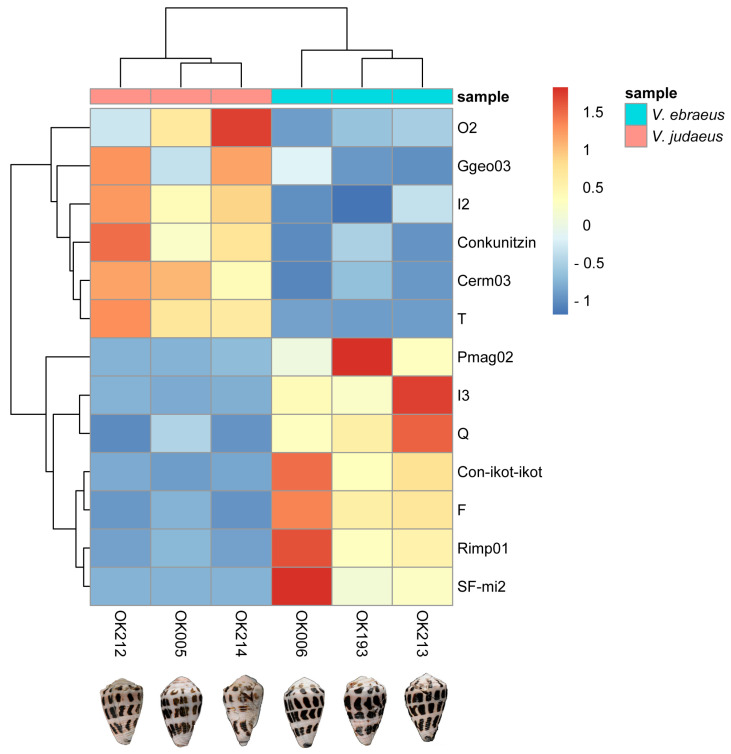
Cluster analysis of differentially expressed superfamilies of conotoxins and hormones between the *V. ebraeus* and *V. judaeus* species. Color gradient from dark blue to dark red, represents the Z-score normalized values of the TPMs.

**Figure 7 marinedrugs-20-00149-f007:**
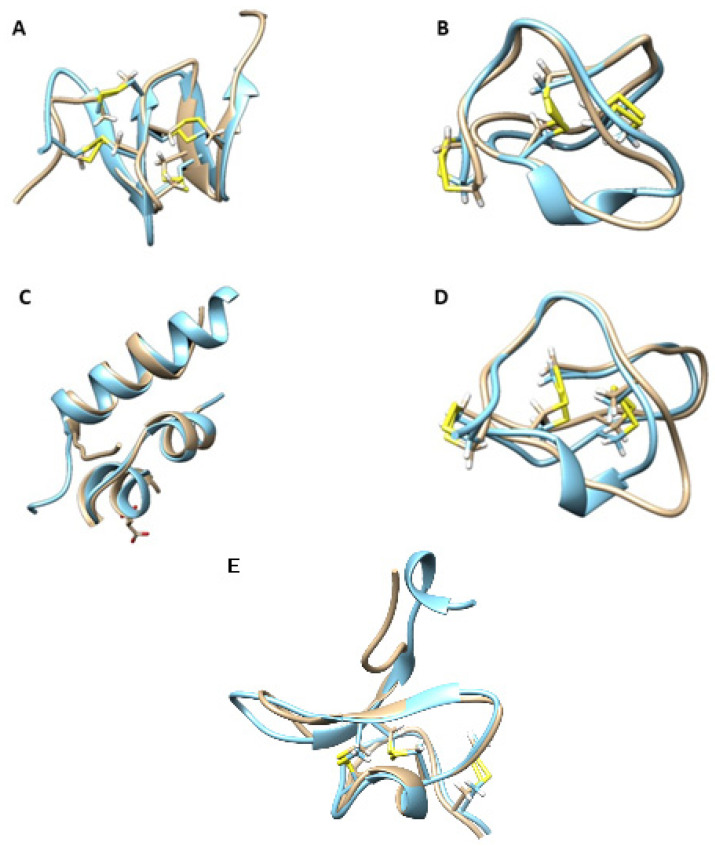
Determination of the 3D structures of relevant venom proteins. Superimposition of the AlphaFold2 model (blue) with known structures (beige). (**A**) OK006_E_244_Sf-mi2 and *Rhizoconus miles* conotoxin ϕ-MiXXVIIA (PDB 6ppc) [24]; (**B**) OK193_E_124_O1 and *Pionoconus magus* ω-conotoxin MVIIA (Ziconotide; PDB 1ttk) [31]; (**C**) OK213_E_290 and *Gastridium geographus* insulin G1 (PDB 5jyq) [32,33]; (**D**) OK160 Eb6.9_O1 and *G. geographus* ω-conotoxin GVIA (PDB 1omc) [34]; and (**E**) Cebr033_J_232_Virro03 and μ-theraphotoxin-Pn3a isolated from venom of the tarantula *Pamphobeteus nigricolor* (PDB 5t4r) [35].

**Table 1 marinedrugs-20-00149-t001:** Venom duct transcriptomes of *Virroconus* analyzed in this work.

Sample Name	Species	Sampling Location	SRA Number	Raw Reads	Clean Reads	Illumina HiSeq	Transcripts	BUSCO (metazoa_odb10; *n* = 954)	Source
OK006	*V. ebraeus*	Okinawa (Japan)	SRR17653519	58,508,016	57,760,881	2500	68,809	C:32.4%[S:25.2%,D:7.2%],F:11.3%,M:56.3%	This work
OK193	*V. ebraeus*	Ishigaki (Japan)	SRR17653518	56,956,142	56,523,495	2500	68,514	C:33.1%[S:25.7%,D:7.4%],F:11.4%,M:55.5%	This work
OK213	*V. ebraeus*	Ishigaki (Japan)	SRR17653517	67,857,352	67,318,628	2500	66,731	C:31.8%[S:24.4%,D:7.4%],F:12.9%,M:55.3%	This work
MAP17	*V. ebraeus*	Bali (Indonesia)	SRR2609538	20,828,802	20,560,135	2000	57,781	C:30.0%[S:25.4%,D:4.6%],F:22.5%,M:47.5%	[15]
CebrOK014	*V. ebraeus*	Okinawa (Japan)	SRR14407590	25,173,828	22,036,629	4000	38,810	C:6.9%[S:5.7%,D:1.2%],F:8.9%,M:84.2%	[14]
CebrOK016	*V. ebraeus*	Okinawa (Japan)	SRR14407576	29,931,016	26,127,506	4000	62,828	C:11.3%[S:9.4%,D:1.9%],F:11.5%,M:77.2%	[14]
OK005	*V. judaeus*	Okinawa (Japan)	SRR17653516	58,301,376	57,429,299	2500	63,846	C:31.6%[S:26.1%,D:5.5%],F:13.5%,M:54.9%	This work
OK212	*V. judaeus*	Ishigaki (Japan)	SRR17653515	79,281,440	78,688,495	2500	95,899	C:45.6%[S:36.1%,D:9.5%],F:14.2%,M:40.2%	This work
OK214	*V. judaeus*	Ishigaki (Japan)	SRR17653514	89,729,936	89,161,848	2500	111,328	C:48.6%[S:37.2%,D:11.4%],F:13.9%,M:37.5%	This work
CebrOK033	*V. judaeus*	Okinawa (Japan)	SRR14407588	33,908,028	30,229,897	4000	82,404	C:25.2%[S:19.3%,D:5.9%],F:20.9%,M:53.9%	[14]
CebrOK035	*V. judaeus*	Okinawa (Japan)	SRR14407589	30,737,140	26,518,858	4000	77,472	C:16.0%[S:13.1%,D:2.9%],F:20.0%,M:64.0%	[14]

**Table 2 marinedrugs-20-00149-t002:** Sequence identification of monoisotopic masses corresponding to main peaks in the UHPLC-MS chromatogram. PTMs are abbreviated as follows: Btr = bromotryptophan; O = hydroxyproline; hVa = hydroxyvaline; Gla = glutamate carboxylation; (Nh2) = N-terminal amidation.

	RT (min)	Peak Mass (Da)	Peptide Sequence
*V. ebraeus*	0.97	3026.162	TCTGNCRLCGAICCCEPKVCRNNQCIDD
	4.98	1619.544	CCKYPLCAAGCSCCTT
	5.03	1440.425	CCC(Btr)RCTRSLH
	5.71	2340.781	CCTFO(hVa)CTACYCCRMHOQHP
	5.76	1620.554	RCCQIVPQCC(Gla)WN
*V. judaeus*	3.66	2226.812	ICPGMCLGGYGK(Gla)PFCHCT(Gla)(Nh2)
	4.32	3318.187	YTONDA(Gla)SS(hVa)CYFLCLMGIDLD(Gla)CNCO(Gla)(Nh2)
	4.85	2571.977	MTLLLEDGCCTRPRCTGACSCCQD
	4.88	2893.113	ECTRSGGACNSHTQCCDDFCSTATSTCI(Nh2)
	5.23	2562.951	LOCCYI(Gla)(Btr)CSRRCICDOLEO(Nh2)
	5.23	2562.951	LOCCYI(Gla)(Btr)CSRRCLCNOLEO
	5.23	2562.951	LOCCYI(Gla)(Btr)CSRRCICNOLEO
	5.38	3256.334	FCONOCQSCOSOG(Gla)C(hVa)RPYTGHTFFLHL
	5.38	3026.162	TCTGNCRLCGAICCCEPKVCRNNQCIDD

## Data Availability

Sequenced raw reads for the newly sequenced transcriptomes were deposited in GenBank under the Bioproject number PRJNA704251. Curated sequences for the new samples of this work (OK006, OK193, OK213, OK005, OK212 and OK214) were deposited in GenBank under accession numbers OM632747–OM634639. Curated sequences from the already published transcriptomes (CebrOK014, CebrOK016, MAP17, CebrOK033 and CebrOK035) were deposited in GenBank separately as third party annotation with the numbers BK059935–BK061130.

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
