# Peer review of "Comparative Venomics of the Cryptic Cone Snail Species Virroconus ebraeus and Virroconus judaeus"

_marinedrugs, 2022, doi:10.3390/md20020149_

Round 1

Reviewer 1 Report

1) Generally, the name of cone snail species is written as C. XXX, the name of V. ebraeus and V. judaeus is confusing, they are should be renamed as C. ebraeus and C. judaeus

2) It is not important to predict the structures of new conotoxins, but it will be helpful to determine the targets of new superfamily conotoxins. I suggest that the authors add these data of two or three new superfamily conotoxins.

3)As for the study on genomics and proteomics, the authors have done a lot of work and compared the reported results, I have not further comments. 

Author Response

RESPONSE TO REVIEWERS

Reviewer #1

1) Generally, the name of cone snail species is written as C. XXX, the name of V. ebraeus and V. judaeus is confusing, they are should be renamed as C. ebraeus and C. judaeus

We follow the classification proposed in Tucker and Tenorio (2009) that divide the family Conidae into different genera, one of them, the genus Virroconus that we used along the manuscript. This classification was based on a comprehensive study of shell, radular morphology, diet habits and other relevant data. However, the traditional single-genus classification was solely based on the morphology of the shell, which is also prone to homoplasy. Moreover, it is very unrealistic (and not very useful) to assign a single genus to a group of more than 1000 species that has been evolving at least during the last 60 millions of years along different habitats and ecological niches. Hence, we would like to maintain our choice of taxonomic classification.

2) It is not important to predict the structures of new conotoxins, but it will be helpful to determine the targets of new superfamily conotoxins. I suggest that the authors add these data of two or three new superfamily conotoxins.

The calculation of the protein structure is an important step to understand the function of a protein because it allows comparison to other proteins of known function.  Following the suggestion of this reviewer, we also tried to infer the structure and function of the new putative conotoxin precursor superfamilies. For two of the superfamilies (Virro01 and Virro02), the structure modeling using AlphaFold rendered non-significant results (average pLDDT < 60). For the sequence CebrOK033_J_232, a member of new superfamily Virro03, a high quality model (average pLDDT = 89) was obtained. The resulting structure was very similar to that of Mu-theraphotoxin-Pn3a from the spider Pamphobeteus nigricolor (Giant blue bloom tarantula), PDBe 5t4r. This toxin is a potent inhibitor of voltage-gated sodium channel NaV1.7. Given the structural resemblances, a similar pharmacological function might be anticipated, although this is yet to be demonstrated. The structure similarity is now shown in the corresponding figure and this results explained in the main text.

3) As for the study on genomics and proteomics, the authors have done a lot of work and compared the reported results, I have not further comments.

We are thankful for this comment and glad to hear that the author appreciates our effort.

References:

Tucker, J. K., & Tenorio, M. J. (2009). Systematic classification of recent and fossil conoidean gastropods: with keys to the genera of cone shells. Conchbooks.

Reviewer 2 Report

The revised manuscript presents a comprehensive analysis of venom composition in two closely-related species of cone snails. The analysis is based on the original data with a good number of replicates. The results of the transcriptomic analysis are backed up by the proteomic data obtained with RP-UHPLC/MS analysis and shotgun proteomics (LC-MS/MS). Although this is a golden standard in exploratory studies of venom composition, unfortunately, not yet consistently followed by researchers.

The analysis is adequate and also follows a well established protocol; I have some technical remarks, but in general I’m satisfied with the quality of the data analysis and data presentation. The figures are clear, and the extensive supplementary material documents the findings quite nicely. Finally, it certainly makes the manuscript more compelling that the authors have identified the toxins with a potential to be developed as drug leads, and have generated additional lines of evidence for their possible targets by modelling their molecular conformations, and comparing those to the pharmacologically relevant conopeptides. Therefore, I find the manuscript a worthy contribution to the knowledge of Conus venom diversity, and recommend it publication.

However, what authors clearly could improve is the clarity of writing. The wording is rough, sometimes inaccurate to details, sometimes just unclear. This seems to be more pronounced in the intro and description of transcriptomic results. The final section of the text, dedicated to the metavenom network looks completely out of place. I agree that the analysis carried out by the authors is interesting and novel, but it does not seem to fit well in this paper. If the diversity and identities of the genes co-expressed with conotoxins is addressed, then it deserves a separate study that would more broadly involve genomic resources (generated for Conus by the same research group), and more extensively use tools of comparative genomics. I would omit it completely. I am attaching the commented PDF file of the manuscrirt, where authors will find some more specific comments.

I am happy to be identified, and I hope, the authors will find this feedback helpful.

Author Response

Reviewer#2

The revised manuscript presents a comprehensive analysis of venom composition in two closely-related species of cone snails. The analysis is based on the original data with a good number of replicates. The results of the transcriptomic analysis are backed up by the proteomic data obtained with RP-UHPLC/MS analysis and shotgun proteomics (LC-MS/MS). Although this is a golden standard in exploratory studies of venom composition, unfortunately, not yet consistently followed by researchers.

The analysis is adequate and also follows a well established protocol; I have some technical remarks, but in general I’m satisfied with the quality of the data analysis and data presentation. The figures are clear, and the extensive supplementary material documents the findings quite nicely. Finally, it certainly makes the manuscript more compelling that the authors have identified the toxins with a potential to be developed as drug leads, and have generated additional lines of evidence for their possible targets by modelling their molecular conformations, and comparing those to the pharmacologically relevant conopeptides. Therefore, I find the manuscript a worthy contribution to the knowledge of Conus venom diversity, and recommend it publication.

We are really encouraged by the nice feedback from this reviewer and would like to thank his/her words.

However, what authors clearly could improve is the clarity of writing. The wording is rough, sometimes inaccurate to details, sometimes just unclear. This seems to be more pronounced in the intro and description of transcriptomic results.

We have revised the wording of the paper and in particular of the introduction and description of transcriptome results following the suggestion of this reviewer.

The final section of the text, dedicated to the metavenom network looks completely out of place. I agree that the analysis carried out by the authors is interesting and novel, but it does not seem to fit well in this paper. If the diversity and identities of the genes co-expressed with conotoxins is addressed, then it deserves a separate study that would more broadly involve genomic resources (generated for Conus by the same research group), and more extensively use tools of comparative genomics. I would omit it completely.

We find interesting to find that some of the proteins associated to venoms in amniotes (the so-called metavenom network) are also present in cone snail venom gland transcriptomes. Hence, we would like to maintain some of the associated results, and following the recommendation of this reviewer, we have deleted the specific section and incorporated these results as part of the Proteome components description.

I am attaching the commented PDF file of the manuscript, where authors will find some more specific comments.

We have incorporated all suggestions provide by this referee.

I am happy to be identified, and I hope, the authors will find this feedback helpful.

Indeed, it is very useful and improves the manuscript.

Reviewer 3 Report

This research article aims to compare the venom of two cryptic species of cone snails using transcriptomics and proteomics. The authors have extracted materials from the venom ducts of several specimen the species of which was identified using mitochondrial bar coding, then used high-throughput mRNA sequencing, then assembled and analyzed contigs, then carried out proteomics analysis or the venom ducts using LC-MS and LC-MS/MS. The authors also modeled some of the peptides discovered in this study using AlphaFold2. The authors conclude that the two tryptic species have different expression levels of gene superfamilies. The conclusion seems supported by the data. This is a well-written manuscript but there are some missing information, which would help reproducibility and clarity of the explanations.

Comment 1. SRA numbers in Table 1 will need to be provided before publication.

Comment 2. In Table 1, there are more "Clean Reads" than "Raw Reads". This does not make sense... please make clear what these number means.

Comment 3. The authors need to describe how was the gene superfamilies were assigned to each conotoxin transcript.

Comment 4. Figure 3B and 3C are insufficiently described. I do not understand what these diagrams represent. Please add some comments in the Figure caption.

Comment 5. The authors are talking about Metavenom Network (MVN) but did not define that term. Could the authors provide a definition as well as a reference?

Comment 6. The use of species name in the name of gene superfamilies, which encompass many if not all cone snail species, is a poor practice. For example Cerm03 was discovered in Conus ermineus but it is also found in other species. Why not using an available letter or add a number to a letter as it has been done previously? For example Virro02 is more similar to the O2 gene superfamily, why not calling it O4 or O5 (O3 being taken)?

Comment 7. What is the biological significance of comparing the cumulative expression level (TPMs) of gene superfamilies? Please provide this information in the text. Comparing the cummulative expression level (TPMs) of cysteine frameworks, which is a proxy for type of activity, would have be more significant. By contrast gene superfamilies represent toxins with a variety of cysteine frameworks (shape), which have different type of molecular targets. For example the A-superfamily toxins target ligand-gated ion channels (nAChRS), voltage-gated ion channel (potassium channels) or GPCRs (adrenoreceptor).

Comment 8. When comparing the TPMs between cryptic species, were the TPMs normalized using house-keeping genes? (ie see lines 292-293 or 302-303). If it was please add this information in the text. If it was not then unfortunately these comparisons are not significant.

Comment 9. Lines 432-438, does the Viro03 gene superfamily peptide displays similar residues to Pn3a for positions that are known to be important for activity? A good SAR has been done on Pn3a. Please provide this analysis in the text.

Comment 10. For the proteomics, I note that the authors state that they considered 5 post-translational modifications ("methionine oxidation, amidation of C-terminus, proline oxidation to pyroglutamic acid, and pyroglutamic from glutamic acid and glutamine"). There are 18 naturally-occurring post-translation modifications recorded in ConoServer, and it would be interesting to add a note that some of them accounts for the masses that could not be identified in the transcriptome. Please provide this information in the text if you agree.

Comment 11. line 246, "signal domain" should be "signal sequence" or "ER signal sequence".

Note. Lines 313-315, I agree that a better nomenclature of conotoxin is needed ... but past experience is that not many groups seem to follow if one is proposed.

Author Response

Response to reviewer 3#

This research article aims to compare the venom of two cryptic species of cone snails using transcriptomics and proteomics. The authors have extracted materials from the venom ducts of several specimen the species of which was identified using mitochondrial bar coding, then used high-throughput mRNA sequencing, then assembled and analyzed contigs, then carried out proteomics analysis or the venom ducts using LC-MS and LC-MS/MS. The authors also modeled some of the peptides discovered in this study using AlphaFold2. The authors conclude that the two tryptic species have different expression levels of gene superfamilies. The conclusion seems supported by the data. This is a well-written manuscript but there are some missing information, which would help reproducibility and clarity of the explanations.

We are grateful for the reviewer comment and for the positive feedback.

Comment 1. SRA numbers in Table 1 will need to be provided before publication.

Now the numbers are correctly provided in the Table 1.

Comment 2. In Table 1, there are more "Clean Reads" than "Raw Reads". This does not make sense... please make clear what these number means.

The number of raw reads corresponded to half of the total sequenced paired-end reads as is normally reported in other papers. However, we agree that this could be misleading when compared to clear reads. Thus, we now report the total pile paired-end reads.

Comment 3. The authors need to describe how was the gene superfamilies were assigned to each conotoxin transcript.

To clarify this point, we now added the following sentence in the material and methods section: “Conotoxin precursor superfamilies corresponded to transcript sequences that showed more than 70% of identity in the signal region.”.

Comment 4. Figure 3B and 3C are insufficiently described. I do not understand what these diagrams represent. Please add some comments in the Figure caption.

We have now completed the caption with the following text “Shared conotoxin precursor and hormone transcripts between of V. ebraeus (blue) and V. judaeus (red). A) Venn plot showing shared transcripts between the six specimens of V. ebraeus and the five samples of V. judaeus. B) Shared transcripts among samples within Samples of V. ebraeus (B) and V.judaeus (C). Colored bars indicate the total number of pre-cursors of each sample, whereas the linked dots indicates the shared transcripts between samples. Y axis indicates number of transcripts.  C) Samples of V. judaeus. The bars represent the different number of transcripts and dots in black are intersects of each bar. Duplicated sequences for each species were collapsed.”

Comment 5. The authors are talking about Metavenom Network (MVN) but did not define that term. Could the authors provide a definition as well as a reference?

We have now added the definition, now it reads “a cluster of housekeeping genes that co-express with venom genes in venomous amniotes” and the reference the first time is mention the MVN.

Comment 6. The use of species name in the name of gene superfamilies, which encompass many if not all cone snail species, is a poor practice. For example Cerm03 was discovered in Conus ermineus but it is also found in other species. Why not using an available letter or add a number to a letter as it has been done previously? For example Virro02 is more similar to the O2 gene superfamily, why not calling it O4 or O5 (O3 being taken)?

The new superfamilies are defined when < 70% of identity is found between signal sequences. This indicates that they are highly divergent compared with known superfamilies and thus, even the most similar result is not necessarily related to the closest known superfamily, as suggested by the reviewer. We prefer to continue with the nomenclature that we use here as it reflects where the original precursor was detected, a practice taken from taxonomy.

Comment 7. What is the biological significance of comparing the cumulative expression level (TPMs) of gene superfamilies? Please provide this information in the text. Comparing the cummulative expression level (TPMs) of cysteine frameworks, which is a proxy for type of activity, would have be more significant. By contrast gene superfamilies represent toxins with a variety of cysteine frameworks (shape), which have different type of molecular targets. For example the A-superfamily toxins target ligand-gated ion channels (nAChRS), voltage-gated ion channel (potassium channels) or GPCRs (adrenoreceptor).

Gene superfamilies reflect evolutionary groups whereas clusters based on cysteine are functional groups. From sequence comparison, it is difficult to assess function which otherwise is prone to convergence. The suggestion of the reviewer could be interesting if function were known for the different conotoxins, but this goes beyond the present work. Instead, we can evaluate the frequency of expression of the different gene superfamilies in an evolutionary context to determine differences among species and diets. We now have clarified this in the main text by adding “The expression of the different conotoxin precursor superfamilies and hormones was compared between V. ebraeus and V. judaeus to find significant differences between these cryptic species and associate them to potential diet differences.”.

Comment 8. When comparing the TPMs between cryptic species, were the TPMs normalized using house-keeping genes? (ie see lines 292-293 or 302-303). If it was please add this information in the text. If it was not then unfortunately these comparisons are not significant.

The suggestion of the reviewer is adequate for quantitative RT-PCR where differences in the final expression need to be normalized by the expression of house-keeping genes used as control. In the case of transcriptomes based on illumine Hiseq reads, the sensibility of the technique is enough to avoid biases during expression quantification. Instead what is important for RNA-seq comparative analysis is to normalize for gene length and sequencing depth, which is the normal practise in the corresponding studies and in our case. See Evans, C., Hardin, J., & Stoebel, D. M. (2018). Selecting between-sample RNA-Seq normalization methods from the perspective of their assumptions. Briefings in bioinformatics, 19(5), 776-792; Bullard, J. H., Purdom, E., Hansen, K. D., & Dudoit, S. (2010). Evaluation of statistical methods for normalization and differential expression in mRNA-Seq experiments. BMC bioinformatics, 11(1), 1-13.

Comment 9. Lines 432-438, does the Viro03 gene superfamily peptide displays similar residues to Pn3a for positions that are known to be important for activity? A good SAR has been done on Pn3a. Please provide this analysis in the text.

We have included the following comment: “In this context, Mueller et al. (2020) have shown that the presence of positively charged aminoacids K22 and K24 was considered crucial for Pn3a activity. In CebrOK033_J_232, these positions are occupied by different aminoacids, namely arginine (R22) and serine (S24). Replacement of K22 by arginine in Pn3a reduces the activity, while position 24 seems more tolerant to modifications. This might indicate a lower activity of CebrOK033_J_232 compared to Pn3a for Nav1.7 inhibition”.

Comment 10. For the proteomics, I note that the authors state that they considered 5 post-translational modifications ("methionine oxidation, amidation of C-terminus, proline oxidation to pyroglutamic acid, and pyroglutamic from glutamic acid and glutamine"). There are 18 naturally-occurring post-translation modifications recorded in ConoServer, and it would be interesting to add a note that some of them accounts for the masses that could not be identified in the transcriptome. Please provide this information in the text if you agree.

This comment has been now added.

Comment 11. line 246, "signal domain" should be "signal sequence" or "ER signal sequence".

Changed as suggested.

Note. Lines 313-315, I agree that a better nomenclature of conotoxin is needed ... but past experience is that not many groups seem to follow if one is proposed.